# A simple model of the attentional blink and its modulation by mental training

**Nadav Amir**[1], **Naftali Tishby**[1,2†], **Israel Nelken**[1,3]*

**1** The Edmond and Lily Safra Center for Brain Sciences, Hebrew University, Jerusalem, Israel, **2** The Rachel and Selim Benin School of Computer Science and Engineering, Hebrew University, Jerusalem, Israel, **3** Department of Neurobiology, Institute for Life Sciences, Hebrew University, Jerusalem, Israel

† Deceased.
* israel@cc.huji.ac.il

## Abstract

The attentional blink (AB) effect is the reduced probability of reporting a second target (T2) that appears shortly after a first one (T1) within a rapidly presented sequence of distractors. The AB effect has been shown to be reduced following intensive mental training in the form of mindfulness meditation, with a corresponding reduction in T1-evoked P3b brain potentials. However, the mechanisms underlying these effects remain unknown. We propose a dynamical-systems model of the AB, in which attentional load is described as the response of a dynamical system to incoming impulse signals. Non-task related mental activity is represented by additive noise modulated by meditation. The model provides a parsimonious computational framework relating behavioral performance, evoked brain potentials and training through the concept of reduced mental noise.

**Data Availability Statement:** All code used for running simulations, parameter fitting, and plotting is available on the OSF repository at https://osf.io/52rmt/.

## Author summary

Mindfulness meditation involves the training of attention and has been shown to improve performance in temporal-attention demanding tasks such as the attentional blink. It allegedly does so by reducing ongoing mental noise in the brain, allowing the practitioner to allocate attentional resources more efficiently. We develop a parsimonious, dynamical-systems based model of the temporal limitations of attention and their improvement through mental training. We show that the model can reproduce the attentional blink effect and explain improved performance following intensive mental training. The model provides a novel, mechanistic account relating the effects of mental training on behavioral performance in the attentional blink task and similar tasks, as well as the associated event related brain potentials.

## Introduction

The temporal dynamics and limitations of attention have attracted considerable interest over the last decades (see [1] for an overview). Whether such limitations can be modified and improved by training is an obviously important question. A number of studies have suggested

**Funding:** This study was supported by Advanced ERC grant 340063 (project RATLAND), F.I.R.S.T. grant 1075/13 to IN and by the Intel ICRI-CI, and the Gatsby Charitable Foundation (NT). https://erc. europa.eu/funding/advanced-grants https://www. isf.org.il/#/ https://www.gatsby.org.uk/ https:// www.intel.com/content/www/us/en/research/ blogs/private-ai-collaborative-research-institute-launch.html The funders had no role in study design, data collection and analysis, decision to publish, or preparation of the manuscript.

**Competing interests:** The authors have declared that no competing interests exist.

that mental training, in the form of mindfulness meditation, can modulate attention allocation [2–7] as well as resting state brain activation patterns [8, 9]. This form of meditation is hypothesized to reduce "mental noise", i.e., non task-related mental activity and mind wandering, which presumably compete with incoming task-related stimuli for limited attentional resources [2, 10]. However, the underlying neural and computational mechanisms for the effects of such mental training on attentional capacity remain unknown.

Here we describe a simple dynamical model of attentional resource allocation. Mental noise is introduced into the model as an additive perturbation that competes with incoming stimuli for attentional resources. We apply the model to a widely studied experimental effect known as the attentional blink (AB) [11, 12], which refers to the reduced ability of subjects to report a second target stimulus (T2) that appears within 200–600 ms after another target (T1) in a rapid serial visual presentation (RSVP) task.

The AB effect has played an important role in studying the limits of human ability to allocate attention over time (see [13, 14] for reviews). It has also been used to study the effects of mental training on cognitive performance and brain resource allocation [2–4]. Of particular interest here is the work of Slagter et al. [2]. The authors compared T2 detection accuracy and T1-elicited P3b amplitudes of 17 meditation practitioners before and after a 3 month meditation retreat using a meditation technique aimed at reducing elaborate object processing. A control group, consisting of 23 meditation novices interested in learning about meditation, received a 1 hour meditation class and were asked to meditate for 20 minutes daily over 1 week prior to each session. The intensive training of the practitioners was associated with improved identification accuracy of T2 and reduced T1-evoked P3b event related potential (ERP) amplitudes, interpreted as evidence for reduced allocation of attentional resources to T1. The present study describes a parsimonious computational model of the temporal dynamics of attentional resource allocation which explains these findings and provides novel predictions.

## Methods

### Model description

In line with previous theoretical accounts of the attentional blink [12, 15], our model postulates that incoming stimuli are processed in two consecutive stages (Fig 1). Both processing stages are modeled as lowpass filters (implemented as finite impulse response filters) with some further processing on their outputs. The lowpass filtering represents the time constants of the neural networks that process the sensory signals.

In the first stage, target stimuli are detected and integrated into a unified sensory signal. We call this signal the *sensory trace* as it postulates a pre-attentive, sensory representation of target identification. The model assumes that at the level of the sensory trace, targets and distractors are perceptually discriminable. This is required in order for the blink to be attentional rather than perceptual. The sensory trace measures the likelihood of a target being present. Since any likelihood measure is obviously bounded by certainty (the value 1), the sensory trace is clipped at unit threshold, making it a sub-linear function of the input signal. Thus, the sensory trace evoked by two temporally close target inputs is smaller than the sum of sensory traces produced by each one of the targets on its own. This sub-linearity is operative when the targets appear consecutively, since the temporal overlap of their sensory representations, and hence the portion of the combined signal which is clipped, will be largest in this case. As we demonstrate below, this component of the model accounts for the lag-1 sparing phenomenon, i.e., the attenuation of the attentional blink when T1 and T2 appear consecutively.

In the second stage, the sensory trace is processed by a limited capacity attentional system for subsequent report or response. Importantly, this stage incorporates an additive noise term

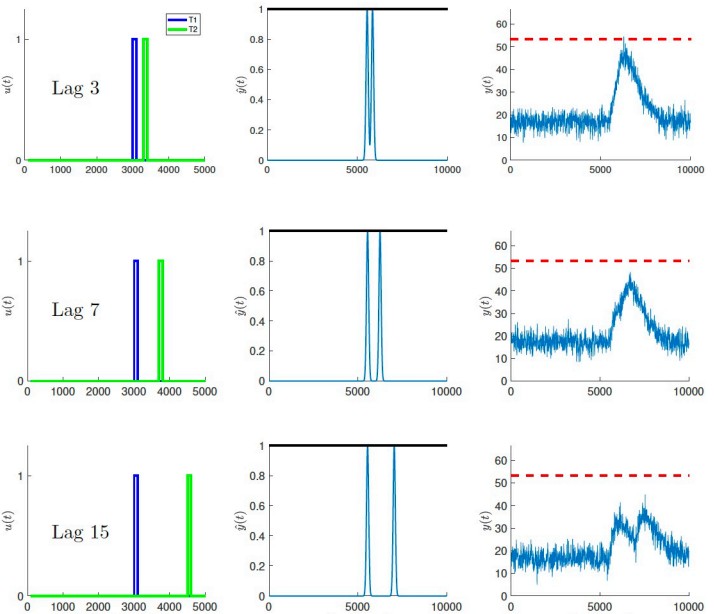

**Fig 1. Attentional blink model.** Input targets (left) are processed in two stages: a unit-clipped sensory trace (center, clipping threshold represented by solid black line) followed by a threshold limited allocation of attentional resources (right, blinking threshold represented by dashed red line). Top: When input targets appear in close temporal succession (here, lag 3) the output of the attentional system is more likely to cross the blinking threshold (dashed red line) resulting in reduced detection of T2 (the attentional blink). Middle: When the temporal interval between targets is longer (here, lag 7), the overlap between their attentional loads is reduced, with a lower chance of crossing the blinking threshold. Bottom: When the two targets are even further apart (lag 15), each one creates a separate peak in the attentional load and the probability of triggering an attentional blink is negligible.

representing ongoing mental noise in the brain. The output of the second stage is referred to as the *attentional load* and measures the amount of attentional resources drawn by the combination of the mental noise and the incoming target stimuli. When the attentional load reaches a predefined limit, called the *blinking threshold*, conscious processing of the signal is assumed to be interrupted, rendering the subject unable to report the identity of any stimulus that elicits a crossing of the blinking threshold. Formally, the sequence of incoming stimuli is described as an impulse train, $u(t)$, indicating the temporal position of the targets:

$$u(t) = \begin{cases} 1 & \text{target on at } t \\ 0 & \text{otherwise} \end{cases}. \tag{1}$$

The temporal resolution of the model is defined by the sampling rate of $\Delta t = 10ms$. Target stimuli are represented by an impulse of 10 samples, corresponding to a duration of $100ms$. Thus, each lag increases the T1-T2 inter-stimulus interval by $100ms$. For example, a T1-T2 lag of 3 corresponds to an inter-stimulus interval of $200ms$ (Fig 1 left column, top). We note that the particular values used for encoding targets and distractors, 1 and 0 respectively, are not essential to the model. If they were encoded by different values, the model would still work by shifting and scaling the signals at later stages of the model. The important point is that targets and distractors are encoded differently.

In the first processing stage, the input signal is filtered with an impulse response function $h_1(t)$, and clipped at a threshold of 1, yielding the *sensory trace*:

$$\hat{y}(t) = \min \begin{cases} u(t) * h_1(t) \\ 1 \end{cases}.$$   (2)

In the second stage, the sensory trace is filtered with another impulse response $h_2(t)$. The second filtering stage represents the time scales of the attentional processes, which are slower than the sensory/perceptual ones. A stationary white-noise Gaussian process, $n(t) \sim \mathcal{N}(\mu, \sigma^2)$, representing ongoing mental noise fluctuations, is added to the filtered signal, yielding the *attentional load*:

$$y(t) = \hat{y}(t) * h_2(t) + n(t).$$   (3)

According to the model, an attentional blink occurs whenever the attentional load crosses a predefined capacity limit, the *blinking threshold*, denoted by $y_B$. The target eliciting the blinking threshold crossing suffers reduced detection. Thus in a typical RSVP trial with two targets, T2 detection may be jeopardized without affecting T1 detection accuracy.

The introduction of the noise term, $n(t)$, is central to the model. It represents ongoing mental noise levels that are postulated to result from mind-wandering and other non-task related mental activity. The noise competes with incoming targets for limited attentional resources. It is specified by its mean, $\mu(t)$, and standard deviation, $\sigma(t)$, representing respectively the baseline and fluctuations levels of the noise. In this paper we use a Gaussian white noise process for the sake of analytical tractability, but the mental noise may have more complex structure such as temporal correlations with a power-law profile [16]. Such correlations may be easily incorporated into the model.

For a given stimulus sequence, behavioral performance level, i.e., blinking probability, is thus determined by the distance between the blinking threshold and the peak attentional load during the response of the system to the input signal. In S1 Text, we derive an analytical approximation for the blinking probability, using the theory of extreme value distributions [17].

To implement the model, specific choices must be made for the filters. While the particular shape of the impulse response function $h_1(t)$ is not essential to the model, its width must be somewhat longer than the inter-stimulus interval in order that two stimuli one lag apart would interfere but two lags apart remain unaffected. We used a Gaussian impulse response with a window length of $L_1 = 90$ samples and a standard deviation of $\tau_1 = 60ms$, corresponding to a half-maximum width of $141ms$.

For the second stage filter, $h_2(t)$ we used the following Gamma function:

$$h_2(t) \propto \begin{cases} te^{-t/\tau_2} & t > 0 \\ 0 & t \leq 0 \end{cases}$$   (4)

with a window length of $L_2 = 263$ samples and a time constant $\tau_2 = 500ms$, which determines the maximum response time of $h_2(t)$. We choose the Gamma function to represent the response profile of the attentional system as it is known to represent well temporal processes related to attention allocation such as pupillary dilation (a correlate of attentional effort) [18] and temporal sensitivity in the visual system [19].

## The P3b brain potential

The P3b is a positive ERP, peaking around 300ms after the triggering event. The amplitude of the P3b is often interpreted as an index of cognitive workload, or of the demand for attentional resources [20, 21]. To model the P3b amplitude, we first defined a *Resource Allocation Index* (RAI): the ratio between the maximal attentional load drawn by a specific stimuli sequence and the total attentional resources available in the system:

$$RAI = \frac{\max_t y(t) - \mu}{y_B - \mu},$$  (5)

where $\mu$ is the mean mental noise level and the maximum is computed over the total duration of the stimulus sequence. The RAI can be considered a measure of attentional resource allocation in the model, with small values ($RAI \approx 0$) indicating low demand on the attentional system, and high values ($RAI \approx 1$) indicating near depletion of attentional resources. We would like therefore to relate the RAI with the P3b amplitude. While the RAI is a dimensionless index, typically taking values between 0 and 1, the range of P3b amplitudes can vary substantially between individual subjects [22]. We suggest here that the P3b measures the RAI, relative to the mental noise fluctuation level of the subject. In other words, to account for inter-subject variability, we posit that the P3b amplitude is the RAI, scaled by the subject-specific standard deviation of the mental noise $\sigma$:

$$P3b = \frac{RAI}{\sigma}.$$  (6)

Mental noise fluctuations thus scale the subject-specific P3b amplitudes. For the same attentional demand level (RAI), larger values of $\sigma$ correspond to smaller T1-evoked P3b magnitudes.

## Summary of model assumptions

To summarize, we proposed a two stage model of attentional resource allocation dynamics. Each stage is implemented by a linear filter followed by some additional processing. The first stage represents the sensory processing of the input target stimuli followed by a clipping non-linearity. The second stage represents the capacity limited attentional channel in which sensory stimuli and internal noise compete for attentional resources. The attentional blink occurs when attentional resources required to process a stimulus are higher than a maximal value. The amplitudes of T1-evoked P3b potentials are hypothesized to reflect the ratio between available and total attentional resources, scaled by a subject specific mental-noise fluctuation parameter (Eqs 5 and 6).

## Parameter estimation

All parameters of the model are summarized in Table 1. The blinking threshold, impulse response parameters and mental noise parameters before training (time 1) were optimized to minimize the squared error between simulated and empirically reported T2 detection rates over different time lags [4]. The values of the optimized parameters were: $\mu_1 = 17.4$, $\sigma_1 = 3.2$, $y_B = 53.3$. The first stage filter was implemented by a Gaussian window, with a standard deviation of $\tau_1 = 60ms$ and a window length of $L_1 = 90$ samples. The second stage filter was implemented by a Gamma function with a time constant of $\tau_2 = 500ms$ and a window length of $L_2 = 263$ samples. The noise parameters after training (time 2) were optimized to fit the reported effects of mental training [2], as explained below in Modeling the effects of mental training. The value of the post-training optimized parameters were: $\mu_2 = 13.2$ and $\sigma_2 = 3.8$.

**Table 1. Model parameters.**

| Parameter name and value | Description |
|---|---|
| $\mu_1 = 17.4, \mu_2 = 13.2$ | Mental noise mean, before and after training |
| $\sigma_1 = 3.2, \sigma_2 = 3.8$ | Mental noise standard deviation, before and after training |
| $y_B = 53.3$ | Blinking threshold |
| $\tau_1 = 60, \tau_2 = 500$ | Time constants (in *ms*) for stage 1 and stage 2 filters |
| $L_1 = 90, L_2 = 263$ | Window length (in samples) for stage 1 and stage 2 filters |

## Results

### Reproducing the attentional blink

We first tested whether the model reproduces the basic AB effect, namely a reduction in detection accuracy of T2 within 200–500 ms after T1 presentation. For a given input stimulus, we define blink occurrence as the event that the attentional load reaches or crosses the blinking threshold during the time period in which the system is responding to the given input signal:

$$P(Blink) = P(\max_t y(t) \geq y_B). \tag{7}$$

We simulated the behavior of the model over $N = 1,000$ repetitions for T1-T2 lags between 1 and 8. We estimated the blinking probability as the proportion of runs in which the attentional load reached or crossed the blinking threshold. We also compared the simulated results with an analytical approximation which yielded similar results (see S1 Text). For the chosen values of the model parameters (see the Parameter estimation section), the model was able to reproduce the typically reported relationship between T2 detection rates and T1-T2 lags (Fig 2).

The increased blink probability for lags 2–4 results from the low-pass filtering of the signal at the first and second stages, which, in turn, causes an overlap between the attentional response of the two targets. Thus, the attentional load of the combined signal is more likely to reach the blinking threshold even in cases where the individual responses to each one are not likely to do so (Fig 1, left). The model results, both simulation and analytical solution, fit well the experimental observations. Note that the model performs better than the data at long lags, suggesting the presence of additional load on the attentional resources that we didn't model here.

### Lag-1 sparing

When T2 appears immediately after T1 (so-called lag-1 occurrence), the overlap between the first stage responses to T1 and T2 is maximal, resulting in a maximal rectification effect due to overflow of the sensory trace above the clipping threshold (Fig 3, center). As a result, the attentional load drawn by the combined response to T1 and T2 in the second stage is reduced for lag-1 compared to the case of lag-2 occurrences (Fig 3, right), with a corresponding reduction in blinking probability, reproducing the "lag-1 sparing" effect. For a given choice of noise parameter values, we computed the probability for attentional blinking, i.e., attentional load crossing the blinking threshold, to occur on trials with a T1-T2 lag of 2, but not on trials with lag 1 nor lag 5 and higher. This calculation was used to constrain the set of possible noise parameters, before and after training, to those which reproduce the typical U-shaped performance curve obtained in empirical studies of the attentional blink. Operationally, we simulated the response of the system 2,000 times for every T1-T2 lag between 1 and 11 and computed the frequency of blinking threshold crossing for each T1-T2 lag. We required that

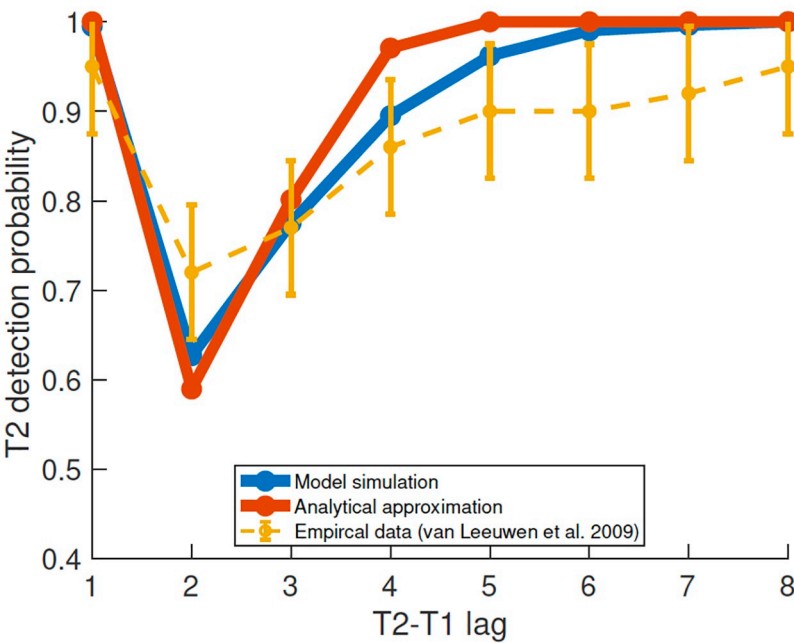

**Fig 2. The attentional blink effect, model and typical data.** T2 detection probability for different T1-T2 lags. Typical empirical data, replotted from [4] (yellow). Model simulated detection probability averaged over $N = 1,000$ repetitions (blue). An analytical calculation of the probability, using double exponential distribution was used to approximate the distribution of the maximal attentional load (red).

the product of the probabilities of the three events: blinking does not occur at lag 1, does occur at lag 2, and does not occur at lag 5 and higher, will be at least 0.2.

## Modeling the effects of mental training

We wanted to capture the effects of mental training in the model through a modulation of the mental noise parameters, which were presumably manipulated through the training procedures in Slagter et al. [2]. Therefore, only the mental noise parameters were modified to model the data after training (time 2). Specifically, the following two effects were reported:

1. Meditation practitioners exhibited a greater reduction in blinking probability between time 1 and 2 compared to novice controls.

2. When comparing T1-elicited P3b amplitudes at times 1 and 2, meditation practitioners exhibited a large amplitude reduction in no-blink trials but not in blink trials. In novices however, P3b amplitudes remained similar at both times for both no-blink and blink trials.

We therefore used the model to estimate behavioral performance and P3b amplitudes for pairs of parameter values, $(\mu_1, \sigma_1)$ and $(\mu_2, \sigma_2)$, representing mental noise statistics before (time 1) and after (time 2) the training period respectively. Following the effects reported by Slagter et al. [2], we defined such pairs to be consistent with the observations when they satisfied all of the following conditions:

1. An increase in T2 detection accuracy, from $0.6 \pm 0.1$ at time 1 to 0.8 or more at time 2, for a T1-T2 lag of 3.

2. A reduction in T1-evoked P3b amplitude between time 1 and 2 by a factor of 1.25–2.

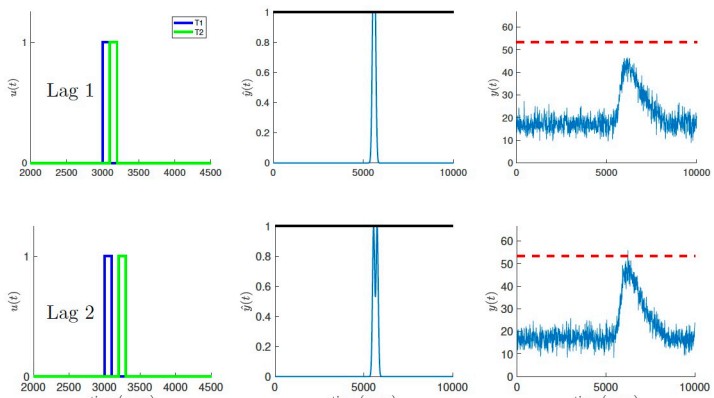

**Fig 3. Lag-1 sparing effect.** When T2 appears immediately after T1 (lag-1, top), their overlap in the first processing stage is maximal, resulting in a stronger reduction of the sensory trace due to the clipping threshold (solid black line). This causes a decrease in the attentional load at the second processing stage (bottom axis) with a corresponding reduction in blinking probability. At lag-2 (bottom), the overlap in the first stage is smaller, resulting in less clipping of the sensory trace and a higher attentional load.

3. Reproduction of the typical U-shape profile of T2 detection probability vs. T1-T2 lag (Fig 3).

Conditions 1 and 2 correspond to the empirical effects 1 and 2 above respectively. Condition 3 was introduced as a further constraint on the range of mental noise parameters, to ensure that they reproduce the characteristic U-shaped performance profile typically observed empirically. As above, this was defined as a requirement that the product of probabilities for attentional blink occurring at lag 2, not occurring at lag 1 and not occurring at lag 5 or higher, would be greater than 0.2.

Fig 4 shows the model-derived values of T2 detection probability (Eq S1 in S1 Text) at a T1-T2 lag of 3 (A), P3b amplitudes (Eq 6) also at a lag of 3 (B), and U-shape profile probability as a function of mental noise parameters (C). Blue and red crosses indicate $(\mu, \sigma)$ parameter pairs which satisfy the conditions 1–3 above (blue crosses correspond to time 1 and red ones to time 2). The model predicts that mental noise baseline level $\mu$ decreases by a factor of about 0.75. Interestingly, the model also predicts that the fluctuations in the mental noise, $\sigma$, increase by about the same amount. For these parameter ranges, the model reproduced the three-way interaction in the T1-evoked P3b amplitudes: a reduction of 30% in T1-evoked P3b amplitude for practitioners versus novices in no-blink versus blink trials at time 2 versus time 1. These results are summarized in Fig 5 (solid bars). The findings of Slagter et al. [2], are also replotted for comparison (lighter hued bars).

The effects of meditation on the attentional load are shown in Fig 6 for a T1-T2 lag of 2. The reduction in mental noise baseline after training (right panel, red trace) reduces blinking probability despite the increase in noise fluctuation.

## Discussion

Our model provides a unified computational account of the attentional blink and the P3b potentials evoked during task performance. Furthermore, modifications of the mental noise component in the model account for both behavioral and electrophysiological effects of mental training reported by Slagter et al. [2]. This noise may be related to mind wandering or thoughts

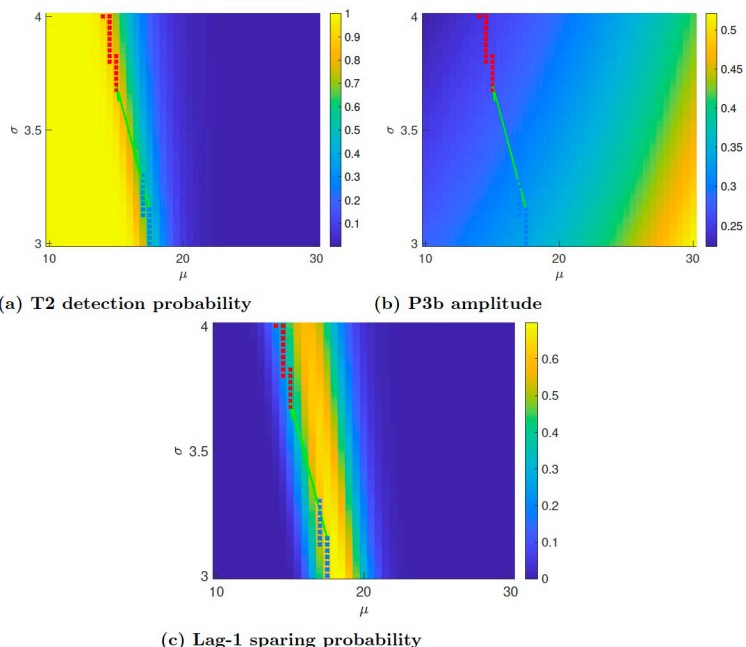

(a) T2 detection probability
(b) P3b amplitude

(c) Lag-1 sparing probability

**Fig 4. Model behavior in mental noise parameter space.** (A) T2 detection probability for a T2-T1 lag of 4. (B) The model P3b amplitude defined in Eq 6, for a T2-T1 lag of 4. (C) lag-1 sparing probability as a function of mental noise activity parameters (mean and variance). The color indicates the probability of crossing the blinking threshold at lag 2 but not at lags 1 and 5. Blue and red crosses indicate $(\mu, \sigma)$ values at time 1 and 2 respectively, for which the model reproduces lag-1 sparing as well as the main findings of Slagter et al. [2], namely an increase of in T2 detection accuracy from 0.6±0.1 at time 1 to 0.8 or higher at time 2, and a reduction in T1-evoked P3b amplitudes by a factor of of 1.25–2. The green arrow connects the pair of mental noise parameter values at time 1 and 2 corresponding to the effects of mental training shown in Fig 5 below.

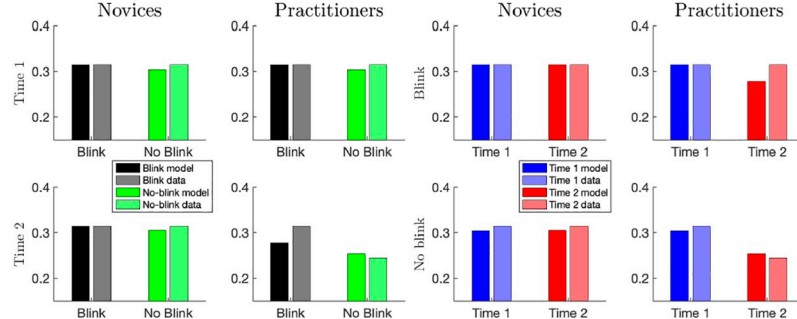

**Fig 5. Simulated and empirical effects of mental training on T1-evoked P3b amplitudes.** Left: T1-evoked P3b amplitude as a function of T2 detection (blink or no-blink), session (time 1 or time 2), and group (practitioners or novices). Meditation practitioners show a greater reduction in T1-evoked P3b amplitude compared to novices in no-blink vs blink trials at time 2 vs time 1. Right: Selective reduction in T1-evoked P3b amplitude in no-blink trials in the practitioner group. Mental noise mean and standard deviation levels $(\mu, \sigma)$, for practitioners: (17.5, 3.15) at time 1 and (14.5, 3.8) at time 2. For novices: (17, 3) at both times. Colors follow figure 3 in Slagter et al. [2], whose data is replotted here.

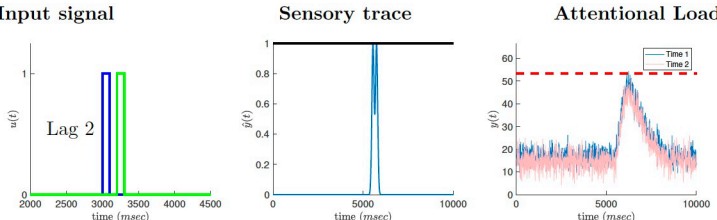

**Fig 6. Mental training induced modulation of attentional load reduces blink probability.** The attentional load profiles before and after meditation (right, blue and red traces respectively). The effects of meditation are modelled as a reduction in mental noise mean from 17.5 to 14.5 and an increase in mental noise standard deviation from 3.15 to 3.8.

that are unrelated to the task, which are known to be reduced following mindfulness meditation [23]. Notably, the model emphasizes the interaction between mental noise and the time scales of attentional processing. An explicit neurophysiologically plausible implementation is not attempted here.

Several computational models for the AB effect have been described (for an overview see the section "Formal Theories" in the review by Dux & Marois [13]). These models can be categorized according to the proposed mechanism hypothesized to underlie the AB. The first type, which includes the Simultaneous Type/Serial Token model [24], suggests that blinking represents an attentional capacity limitation due to T1 processing. The second type, represented by the Boost and Bounce model [25], proposes top-down inhibition of attention due to distractors following T1 as the mechanism responsible for attentional blinking. Our model is similar in spirit to the first type. Its simple and transparent structure makes it particularly easy to understand and analyze. More specifically, the model accounts for changes in T2 detection accuracy at different inter-target intervals, relating them to T1-evoked P3b amplitudes and mental noise levels. The model suggests that the increase in T2 detection accuracy, as well as the mental training induced decrease in T1-evoked P3b amplitudes reported by Slagter et al. [2], can be explained as resulting from a combined reduction in the baseline and increase in the size of the fluctuations of the mental noise. The model also posits that early sensory processing is subjected to a saturation threshold, resulting in sub-linear depletion of attentional resources when targets appear in close temporal proximity. This threshold mechanism accounts for the "lag-1 sparing" effect: the somewhat surprising finding that attentional blinking is often attenuated when T2 appears directly after T1 [26].

The early sensory integration stage of the model suggests that visual stimuli separated by a brief interval are integrated into a unified precept. Such temporal integration effects are well documented and have been hypothesized to result from the temporal overlap between the activity elicited by two brief sequential visual stimuli [27]. Importantly, our model suggests that such integration of temporally proximal targets results in sub-linear activation of sensory representations, implemented in the model by the clipping threshold applied at the sensory integration processing stage. This hypothetical threshold mechanism implies that the activation induced by temporally proximal sequential stimuli would be lower, and with shorter duration, than the combined activation of each stimuli alone. Interestingly, sub-linear integration mechanisms have previously been implied in 'max' operations [28] and orientation selectivity [30] in the primary visual cortex. Our model suggests that a similar mechanism could also play a role in enhancing the resolution of temporal attention for brief consecutive sequences of visual targets.

A related prediction of the model is that adjacent target stimuli are merged into a single, undifferentiated signal at the integration stage, thereby possibly losing information about the unique identity of each one. This is consistent with the finding that when T2 appears immediately after T1 (lag 1 trials), report order is often reversed [15, 30].

The role of distractors in generating the attentional blink is a matter of ongoing debate. Under certain accounts, the attentional blink is caused by the appearance of a distractor after T1, triggering a disruption in attentional control [25, 31]. These accounts are in line with the view of the P3b potential as reflecting context updating [22], in this case switching from distractor to target processing. Another family of accounts attributes the attentional blink to resource capacity limitations [15, 32]. Our model supports such capacity limitation accounts, suggesting that attentional resource consumption is determined by the interstimulus interval. We assume that targets are identified and segregated from distractors at the pre-attentional sensory stage. In consequence, distractors do not draw attentional resources, and the P3b amplitudes reflect attentional resource allocation rather than a context update.

Our model is unique in providing a parsimonious account of the effects of mental training on attentional capacity and the associated event related brain potentials. Importantly, both sets of results were based on the properties of a single internal signal, the overall attentional resources engaged during task performance. The overall amount of attentional resources in the model is composed of two components: those resources that are engaged by the task, and a mental noise that accounts for all task-independent resources that are used, reflecting the presumed mind wandering. We modeled the effect of meditation training as changes in the properties of this mental noise. As expected, the model predicts that the baseline mental noise level is reduced following intensive mental training. Unexpectedly, the model also suggests that the fluctuations of the mental noise, as captured by its standard deviation, increase following mental training (Fig 4). The predicted decrease in mean noise levels is consistent with the reported association of meditation with decreased mind wandering and reduced default mode network activity levels [9, 10]. The increased fluctuations in mental noise following meditation, predicted by the model, may appear surprising, but this prediction falls in line with recent studies suggesting that mental states induced by meditation, as well as by psychedelic substances, are characterized by increased variability in brain oscillatory behavior and signal diversity measures [33–35].

The goal of mindfulness meditation is often described as becoming aware of ongoing mental activity and changing one's attitude towards it, rather than actively manipulating it [36]. Indeed, Slagter et al. interpreted the effects of meditation on the attentional blink as a top-down regulation of the engagement with the sensory trace in response to the first target [2]. In its current formulation, our model does not account for such meditation-specific effects on the sensory trace. Rather, it implements a task-unrelated modulation of mental noise in the attentional processing stage which modulates behavioral performance and P3b amplitudes. Admittedly, while mindfulness meditation may reduce mental noise, it does not necessarily do so. Conversely, mental noise can be reduced in other ways as well. The model predicts that changes in mental noise would affect attentional performance and P3b amplitudes in a similar way to meditation.

In a more recent study, van Vugt et al. compared attentional blink performance in expert meditators performing two different types of meditation: *focused attention*, in which attention is focused tightly on an object, and *open monitoring*, in which one is simply aware of whatever comes into awareness [5]. This study reported a smaller attentional blink during open monitoring compared to focused attention meditation, indicating that particular meditative states can also influence the magnitude of the attentional blink. While that finding was also interpreted in terms of reduced top-down processing of T1, our model provides an alternative

explanation, namely that open monitoring meditation causes a larger reduction in mental noise compared to focused attention meditation.

## Supporting information

**S1 Text. Analytical approximation of blinking probability.**
(PDF)

## Acknowledgments

We thank Leon Deouell and Tamar Regev for helpful discussions.

## Author Contributions

**Conceptualization:** Nadav Amir, Naftali Tishby, Israel Nelken.

**Formal analysis:** Nadav Amir, Israel Nelken.

**Funding acquisition:** Naftali Tishby, Israel Nelken.

**Methodology:** Nadav Amir, Israel Nelken.

**Supervision:** Naftali Tishby, Israel Nelken.

**Visualization:** Nadav Amir, Israel Nelken.

**Writing – original draft:** Nadav Amir, Israel Nelken.

**Writing – review & editing:** Nadav Amir, Israel Nelken.

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
