## [Decision Letter · Decision Letter 0]

29 Mar 2022

Dear Dr. Nelken,

Thank you very much for submitting your manuscript "A simple model of Attentional Blink" for consideration at PLOS Computational Biology.

As with all papers reviewed by the journal, your manuscript was reviewed by members of the editorial board and by several independent reviewers. In light of the reviews (below this email), we would like to invite the resubmission of a significantly-revised version that takes into account the reviewers' comments.

The reviewers are generally enthusiastic about your manuscript, but have some good suggestions (I think) for clarifying the theoretical background and implications of the model. I encourage you to follow these recommendations and submit a substantial revision of your manuscript.

We cannot make any decision about publication until we have seen the revised manuscript and your response to the reviewers' comments. Your revised manuscript is also likely to be sent to reviewers for further evaluation.

Sincerely,

Marieke Karlijn van Vugt, PhD

Associate Editor

PLOS Computational Biology

Samuel Gershman

Deputy Editor

PLOS Computational Biology

Reviewer's Responses to Questions

**Comments to the Authors:**

Reviewer #1: The authors present a dynamical-systems model of the AB, suggesting that meditation-induced modulations as previously observed in the pattern of attentional behavior as well as ERP responses are due to the interplay between attentional load and mental noise levels over time. In addition to demonstrating these effects in the context of the AB, the presented model can also account for lag-1 sparing and the spreading of lag-1 sparing.

General evaluation: The manuscript is well-written and presents an elegant model, accounting for several aspects of the AB phenomenon. The embedding with the existing literature is also done well. I lack the mathematical background to thoroughly judge the soundness of the model’s approach, but the underlying mechanism is generally well explained. I found the model’s surprising prediction that mental noise fluctuations would be increased following intense meditation of particular interest, but also found the ‘clipping of the sensory trace’ an intriguing and plausible mechanistic account of lag-1 sparing. I only have a couple of minor points as detailed below.

Minor points:

Fig 3: The clipping threshold is supposed to be represented by a solid red line, which is missing in the figure. I assume the solid black line was meant to be red?

P9, line 220: please add ‘amplitude’ before ‘reduction’, so that it becomes clearer that the word ‘reduction’ refers to P3b amplitudes rather than a ‘reduction in no-blink trials but not in blink trials’.

Fig 6 is rather small, the colors and crosses are rather hard to distinguish.

P 10, line 250: finding -> findings

Figure 4: baseline activity in attentional load seems to be lower than in figure 3. Why?

The authors write on page 11 that observed changes in behavior and P3b amplitude “can be explained as resulting from a combined reduction in the baseline and increase in the size of the fluctuations of the the mental noise”. It would be useful to see what baseline is referred to here. Could it, along with the effects of meditation on the baseline, perhaps be visualized in a figure similar to Figures 1, 3 and 4?

P 11, line 273: the the mental noise -> the mental noise

Perhaps the title of the manuscript is literally a little too simple. A slightly less minimalistic title might better capture the attention of the reader and cover more of what the paper has to offer.

Reviewer #2: Amir and colleagues proposed in their manuscript a single model of attention blink. Their model assumes that the blinking reflects an attentional capacity limitation due to T1 processing. Their model aimed to provide an account for the reduction of the AB effect following intensive mental training in the form of mindfulness meditation, with a corresponding reduction in T1-evoked P3b brain potentials. The concept of mental noise and its reduction by meditation training is central in the model.

The model is timely because there are still very little modelling efforts in the meditation literature. The strength of this work is that it reproduces with a parsimonious computational model both the behavioral performance, evoked brain potentials and training effect found in Slagter and colleagues. The modeling part and the simulations look adequate from what I can tell. The presentation of the literature and the results are satisfactory. The main limitation of this work is that it is not specific enough to model the regulatory mechanisms of mindfulness practices. This limitation somewhat reduces the explanatory power of this model. The model is ultimately about the role of mental noise on the AB rather than on the effects of mindfulness meditation on AB. The model does not explicitly model the top-down effects of meditation on the sensory trace. These limitations need to be highlighted and discussed in the text and in the manuscript in general.

More specifically,

While I agree with the idea that mindfulness meditation can lead to a reduction of mental noise (e.g. work on mind-wandering and meditation), meditation training might not necessary reduce mental noise and the reduction of mental noise might not be specific to meditation. More specifically, the meta-cognitive stance of mindfulness does not aim at blocking thoughts and emotions, but rather at becoming aware of them, to monitor them without rejecting them (e.g. see the literature on depression and mindfulness, e.g. Teasdale et al). So the goal of the practice is not directly to reduce mental noise but one’s relationship to mental noise. Conversely, there are many activities which could reduce mental noise (walking in nature, being absorbed in some sportive active activity (i.e. concentrative exercise). These activities would not technically qualified as a mindfulness exercise. So this model of AB might not be helpful to understand the mechanism underlying Slagter et al.’ findings.

In line with this idea, Van Vugt and Slagter reported in another study, a differential effect of two styles of meditation on the AB within the same participants: open monitoring style of mindfulness reduced more the AB than a focused style of mindfulness in expert meditators. The baseline level of mental noise during the performance cannot alone explain the behavioral differences found in these two meditative states (because the current model will behave similarly during each meditative state, and therefore I would think that this model can not explain the data from this other study).

This finding needs to be discussed in the discussion as a current limitation of the model.

Following this point, the current model posits that the regulation of the AB effect following the retreat is due to a reduction of mental noise, while Slater and colleagues interpreted their findings rather as produced by a cognitive stance modulating the engagement in the sensory trace itself in response to T1 stimulus. (here is the initial quote: “In this common style of meditation, one starts by focusing or stabilizing concentration on an object such as the breath. Then one broadens one’s focus, cultivating a non-reactive form of sensory awareness or ‘‘bare’’ attention. This form of attention is non-reactive in the sense that, ideally one does not become caught up in judgments and affective responses about sensory or mental stimuli.”.

The authors need to state more clearly to the readers that their model is not meant to model the regulatory effect of mindfulness as proposed by Slagter et al. In the current formulation, the model cannot account for a top-down influence on the sensory trace specific to the meditation. It is rather a task unrelated lack of top-down influence (i.e. the mental noise), which penalizes the magnitude of the P3b (Mental noise fluctuations scales the subject-specific P3b amplitudes line 136).

I was surprised that the authors did not model explicitly the non-target visual stimuli (visual letters in Slagter et al). They suggested at the end of the text that it could be added. It would have made their model more physiologically plausible as the P3b is often viewed as reflecting context-updating (Polich, 2003) (without visual letter there are no context updating..). In this view, stimuli enter the processing system and a memory comparison process is engaged that ascertains whether the current stimulus is either the same as the previous stimulus or not. If the incoming stimulus is not the same and the subject allocates attentional resources to the target, the neural representation of the stimulus environment is changed or updated, such that a P300 (P3b) potential is generated in addition to the sensory evoked potentials. I would encourage the authors to do it, but it is not a requirement. At least the authors need to discuss whether they think their second stage model is neurophysiological plausible or whether it is just described the slow time scale of attentional processes.

**Have the authors made all data and (if applicable) computational code underlying the findings in their manuscript fully available?**

Reviewer #1: Yes

Reviewer #2: Yes

PLOS authors have the option to publish the peer review history of their article (what does this mean?). If published, this will include your full peer review and any attached files.

Reviewer #1: **Yes: **Sander Martens

Reviewer #2: **Yes: **Antoine Lutz
---

## [Decision Letter · Decision Letter 1]

17 Jul 2022

Dear Dr. Nelken,

We are pleased to inform you that your manuscript 'A simple model of the attentional blink and its modulation by mental training' has been provisionally accepted for publication in PLOS Computational Biology.

Best regards,

Marieke Karlijn van Vugt, PhD

Associate Editor

PLOS Computational Biology

Samuel Gershman

Deputy Editor

PLOS Computational Biology

Both reviewers are happy with your revised manuscript, and I also think it has become a very nice paper. Congratulations! I would recommend you check the figures, as they did not show up in the compiled PDF, and also as one of the reviewers indicated, there may have been a switch between Figures 6 and 7. Otherwise, great work!

Reviewer's Responses to Questions

**Comments to the Authors:**

Reviewer #1: I'm happy with the changes the authors made in response to the reviews (assuming the mentioned 'Figure 7' is actually Figure 6.

Reviewer #2: The authors have adequately addressed my questions in their revised manuscript.

Congratulation for this interesting model.

**Have the authors made all data and (if applicable) computational code underlying the findings in their manuscript fully available?**

Reviewer #1: Yes

Reviewer #2: Yes

PLOS authors have the option to publish the peer review history of their article (what does this mean?). If published, this will include your full peer review and any attached files.

Reviewer #1: No

Reviewer #2: **Yes: **Antoine Lutz

---

## [Editor Report · Acceptance letter]

23 Aug 2022

PCOMPBIOL-D-22-00193R1 

A simple model of the attentional blink and its modulation by mental training

Dear Dr Nelken,

I am pleased to inform you that your manuscript has been formally accepted for publication in PLOS Computational Biology. Your manuscript is now with our production department and you will be notified of the publication date in due course.

With kind regards,

Zsofia Freund
